# Repeatability of wildlife surveys for estimating abundance: A method to assess the consistency of detection probability and animal availability

**David M. Delaney**[1]*, **Tyler M. Harms**[2], **Jonathan P. Harris**[1], **Dan J. Kaminski**[2], **Jace R. Elliott**[2], **Stephen J. Dinsmore**[1]

**1** Department of Natural Resource Ecology and Management, Iowa State University, Ames, Iowa, United States of America, **2** Iowa Department of Natural Resources, Boone Research Station, Boone, Iowa, United States of America

* dmdelane@iastate.edu

## Abstract

A primary consideration of abundance studies that use unmarked animals is whether survey counts accurately reflect the population size or if unknown variation in animal movement or detection probability biases counts irrespective of population size. We posited that high repeatability in counts among temporally replicated surveys would indicate that counts are a good index of abundance. We temporally replicated 49 nocturnal spotlight surveys of white-tailed deer (*Odocoileus virginianus*) up to three times each (*n* = 128 total samples) to test the repeatability of this commonly used wildlife monitoring technique. Repeatability was high (*R* = 0.86), suggesting spring spotlight surveys provide a reliable index of deer population size in Iowa, USA. Fourteen percent of the variation among replicated counts was explained by day of year and, to a lesser degree, a vegetation green-up index. Detection probability was high (~0.70) early in the sampling season and declined considerably during the following 6 weeks. Deer abundance was greater at sites with higher percent landcovers of forest and hay/pasture and was lower at sites with higher landcover in crops. Our findings suggest deer managers should sample prior to green-up in the spring to maximize the proportion of the population that is detectable, and that accounting for seasonality on detection estimation is important for reliable abundance estimates if sampling occurs over a range of phenological progression. Finally, we show that temporal replication of surveys is a logistically feasible method to assess the reliability of abundance estimates from study designs that are normally conducted with single visits.

## Introduction

Abundance is an important parameter of interest to ecologists and natural resource managers. Estimates of abundance are used to set harvest quotas, assess conservation practices, and test ecological theory [1]. Techniques to estimate abundance from unmarked animals are useful because they require less effort than mark-recapture techniques [2], and can therefore be implemented over a larger area per unit time [3,4]. Line and point transects are commonly

**Data availability statement:** All relevant data are within the manuscript and its Supporting Information files.

**Funding:** The author(s) received no specific funding for this work.

implemented techniques to count animals [5,6], which can be paired with N-mixture models [7,8] or distance sampling [9–11] to estimate abundance when detection probability is imperfect. In lieu of estimating detection probability, the number of animals observed per unit effort can be used as an index of relative abundance [4,12]. For all of these techniques, meaningful estimates of abundance require that a survey count be representative of the population at the spatial and temporal extent of the estimate. In particular, variation in either animal density or detectability across space or time can affect the number of animals counted irrespective of population size [13–16].

The reliability of survey counts depends on the homogeneity or predictability of two main components that determine how many animals are counted: (1) availability of animals to sample, and (2) probability of detection given availability (*sensu* [14]). N-mixture and distance sampling models both estimate the abundance of the population that is available to sample, and abundance will be underestimated by the proportion of the population that is unavailable (e.g., availability bias [17–19]). Availability bias can be reduced by conducting surveys when animals are maximally exposed (e.g., early morning point counts for passerines, [20]) or mathematically accounted for, if quantified, to improve the accuracy of abundance estimates [17,21]. Additionally, N-mixture and distance sampling models estimate the probability of detection of the available animals, which may be affected by a variety of factors including observer [22], weather [23], or land cover [23] and cause variation in counts. If availability and detectability are homogeneous over space and time, temporally replicated surveys should produce similar counts and serve as a reliable index of population size (e.g., [24]). Confidence in survey counts can also be gained by identifying and modeling factors that explain variation among replicate counts, which enables accurate estimation of abundance (e.g., [14,25,26]). For systems with repeatable counts and predictable detection probability, researchers can reduce visits to one sample per site and reallocate effort elsewhere.

Nocturnal spotlight surveys are commonly used to estimate an index of population size for a variety of wildlife [27–33], but have been criticized as being inaccurate because detection probability can vary unpredictably (reviewed in [34]) and be lower than other methods (e.g., thermal imaging, [28]; aerial surveys, [35]). For species that feed in open habitat at night compared to denser diurnal bedding habitat, conducting nocturnal surveys can increase the proportion of the population that is available to sample, and eyeshine (i.e., light reflection off the tapetum lucidum) can improve detectability [36–38]. Availability is rarely estimated during spotlight surveys and reports of high variability in detection could be a combination of availability and detection [19]. For example, spotlight surveys of white-tailed deer (*Odocoileus virginianus*) in forested regions report high and unpredictable variation in detection probability [27], which is attributed to variation within and among observers, temporal replicate, and spatial location [31,34]. Despite criticism, similar spotlight surveys have been deemed reliable in estimating relative abundance of red deer (*Cervus elaphus*) in forested regions of Italy [39] and France [25] and white-tailed deer in agricultural landscapes of Iowa, USA [40,41]. Regardless of concerns, spotlight surveys are likely to remain a widely used wildlife monitoring technique because of their balance of cost, labor, and assumed accuracy [42]. Given this, robust assessment of potential bias is recommended to improve the accuracy of abundance estimation and increase the usefulness of survey counts [43].

We were interested in whether nocturnal spotlight survey counts of white-tailed deer in Iowa were repeatable from late March to early May when the Iowa Department of Natural Resources conducts annual population monitoring [40,44]. Staff traditionally drive approximately 78 km of gravel roads in each of Iowa's 99 counties with no temporal replication. Our goal was to temporally replicate surveys and assess whether one sample per county is adequate or whether future efforts should trade off such broad spatial monitoring for increased effort

per unit area. We hypothesized that if availability and detection probability were stable during this period of spring, temporal replicates would produce consistent counts, suggesting single surveys would provide adequate population indices. We posited that variation in replicate counts resulting from observation-level covariates could be analytically corrected to produce more accurate estimates of population size. In contrast, poor consistency in counts among temporal replicates that was not explained by covariates would suggest spotlight surveys do not produce meaningful counts during this period and alternative sampling techniques should be explored.

## Study area

The Iowa Department of Natural Resources conducts nocturnal spotlight surveys of white-tailed deer each spring in each county to estimate an index of population size [40,44,45]. These surveys typically occur between mid-March and mid-May. Early during this time period, food is limited and deer frequently feed in open habitat [46–50] and in larger groups [51]. However, constraints on staff time and local weather conditions often cause some counties to be sampled later when spring green-up has occurred to varying degrees.

Surveys for this study were conducted along gravel roads in 18 counties of Iowa, USA (Fig 1). The landcover within 400 m buffers of county-level transects ranged from 1–25% forest, 2–48% hay/pasture, and 27–84% row crop agriculture (data from U.S. Geological Survey National Land Cover Database (NLCD) [52]). The average quality of deer habitat (quantified with a resource selection function from [40]) within 300 m of transects was highly correlated with the average quality of habitat of the broader county they represent (correlation coefficient = 0.97, unpublished data) which is attributable to high spatial coverage (78 km of transect route per county on average) and consideration of available habitat during route selection. Detailed description of the flora, fauna, and landscape of Iowa, as they pertain to deer abundance, occur in Elliott and Harms [53].

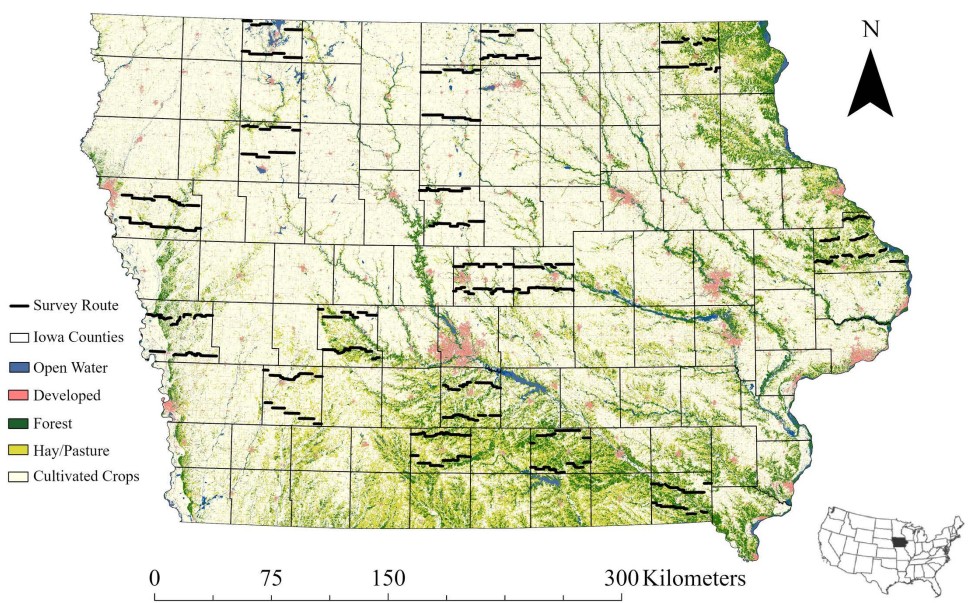

**Fig 1. Map of study locations in Iowa, USA.** Bolded black lines indicate transect routes for the nocturnal spotlight survey used to sample populations of white-tailed deer (*Odocoileus virginianus*). Base map data provided by [54].

## Methods

To assess the repeatability of survey counts, we temporally replicated surveys in 18 counties up to three times per year for up to three years (2017, 2018, and 2023). Intervals between samples of the same site within years averaged 7.7 days (range 1–28 days). We assumed the sample populations were closed between temporal replicates because no hunting season or birth occurred during this time. Our dataset therefore consisted of 128 samples of 49 populations we assumed were closed.

Surveys began one hour after sunset and were restricted to nights with temperature > 0° C, wind < 24 km/h, relative humidity > 40%, and no rain or fog. Two observers drove an average of 41 km (± 6 SD, range = 30–59) of gravel roads in each county at less than 32 km/h, each scanning with a spotlight on one side of the road. Observers recorded relative humidity (%) and temperature (°C) at the start and end of each survey. These variables were used later as observation covariates in the analysis. We summed the total number of deer observed each survey.

We used ArcGIS Pro (versions 3.3.0) to quantify the percent of each landcover type within 400 m of the transects driven in each county (2021 NLCD, 30-m resolution [52]). We focused on forest, hay/pasture, and crop landcover as they were the dominate cover types in rural Iowa. We quantified the variation in elevation within 400 m of each transect by calculating the standard deviation of the elevation of 30 m raster cells from a digital elevation model [55].

### Permits

We collected data by observing animals from publicly owned roads, and animals were not captured or marked. Therefore, our study did not require field site permission or Institutional Animal Care and Use Committee approval.

### Analysis

We analyzed data in the R statistical environment (version 4.2.2). We first examined correlations among observation- and site-level covariates using the package *GGally* [56] to determine if we would separate predictors into multiple models to avoid collinearity.

To quantify the repeatability (*R*) of survey counts, we used linear mixed models in the package *rptR* [57] with site–year as a random effect (i.e., "site–year" was fit as a single term in our model, which fit a random intercept for each site–year combination in the dataset). We calculated both global repeatability (fraction of total variance explained by site–year) and adjusted repeatability (repeatability after accounting for the fixed effects of temperature, humidity, and day of year (*sensu* [58]). Repeatability ranges from 0 to 1, and a value of 1 would indicate that all variation is explained by site–year. We modeled the number of deer observed during each survey as a response variable with a Gaussian error distribution, which was a better fit than a Poisson distribution (Δ Akaike information criterion (AIC) = 252).

**N-mixture model framework.** To test whether a variety of observation- and site-level covariates relate to detection and abundance, we built a series of N-mixture models in the *unmarked* package [59]. Because of covariation among potential predictors, limited sample size, and potential global model complexity, we built N-mixture models in four steps to identify final models: 1) identification of observation-level variables on detection, 2) comparison between day of year and NDVI on detection, 3) addition of site-level variables on detection, and 4) addition of site-level variables on abundance.

N-mixture models estimate detection probability by assessing the variation in counts among temporally replicated surveys and estimate abundance based jointly on the variation in counts among sites and average detection probability. The effect of a single predictor can

be simultaneously estimated on both detection and abundance because the N-mixture model derives abundance and detection probability effects from these different sources of variation in counts [60]. As is the case for most N-mixture models [19,61,62], we were not able to separate detection from availability. We modeled abundance with a negative binomial distribution (c-hat = 1.58) because Poisson ($\Delta$ AIC = 2778) and zero-inflated Poisson ($\Delta$ AIC = 2780) yielded poorer model fits for intercepts-only models in an *a priori* analysis.

**Observation-level variables on detection.** We included temperature, humidity, day of year, and year (as a factor) as predictors of detection probability and fit an intercept on abundance. We also included an interaction of day of year and year because sampling began and concluded at different times each year in response to variation in vegetation green-up phenology among years. We used the dredge function in the package *MuMIn* [63] to fit all possible models that varied in the combination of predictors. We determined the best model as the model with the lowest AIC [64,65] and did not consider models with uninformative parameters (i.e., predictors that did not have a significant relationship with the response) [66].

**Comparison between day of year and NDVI on detection.** Based on results from the prior model, we also calculated an index of vegetation green-up (Normalized Difference Vegetation Index, NDVI, S1 Appendix). To do this, we used NDVI data available from the National Oceanic and Atmospheric Administration Climate Data Record [67], which was correlated with day of year ($r$ = 0.30 in global dataset; $r$ = 0.41–0.47 within year subsets, S1–S4 Figs). We built two additional models to test how green-up compared with day of year in predicting detection probability: (1) a model with NDVI and year as predictors and (2) a model with an interaction of NDVI and year as predictors.

**Addition of site-level variables on detection.** To test whether site-level variables covaried with detection probability, we selected the prior top model and added the following predictors: elevation variation, forest percent, hay/pasture percent, and crop percent. Each additional predictor was fit in a separate model because of high correlation among these variables (S5 Fig).

**Addition of site-level variables on abundance.** To test whether site-level variables covaried with abundance, we selected the prior top model for detection probability and ran a series of models including the addition of elevation variation, forest percent, hay/pasture percent, and crop percent as predictors on abundance. Again, each of these predictors was run in separate models because of correlation among these variables. Each model of abundance included transect length as a covariate to account for variable sampling effort among sites. We considered the model with the lowest AIC that included covariates on both detection probability and abundance as our top model.

## Results

### Repeatability

Survey counts were very repeatable, both globally ($R$ = 0.86 [95% CI = 0.78–0.92]; S6 Fig) and after accounting for other potential sources of variation ($R$ = 0.88 [95% CI = 0.81–0.93]; S7 Fig). Accounting for observation-level covariates reduced the unexplained variance among temporal replicates (i.e., the within site–year variance) by 14% (from 911 [95% CI = 648–1193] to 788 [CI = 551–1038]).

### Detection

Detection probability declined with increasing day of year and the magnitude of this effect varied among years (Table 1; Fig 2A; S1 Table). Detection probability also declined with

**Table 1. Results from the best fit N-mixture model assessing the detection probability and abundance of white-tailed (*Odocoileus virginianus*) deer during nocturnal spotlight surveys in Iowa, USA.** Coefficients for year are compared to 2017. Hay/pasture percent was highly correlated with forest ($r = 0.83$) and crop percent ($r = -0.97$) and each are also significant predictors of abundance when included in place of hay/pasture percent (forest $\beta = 3.9$, CI = 1.6–6.2; crop $\beta = -1.5$, CI = −2.2−−0.7). Dispersion estimate = 1.5 ± 0.2 SE ($z = 7.3$, $p < 0.001$).

|  | Estimate | lower 95% CI | upper 95% CI | P |
|---|---|---|---|---|
| **Abundance (log-scale):** | | | | |
| intercept | 5.36 | 5.22 | 5.50 | 0.000 |
| transect length | −0.02 | −0.04 | 0.01 | 0.162 |
| hay/pasture percent | 2.32 | 1.30 | 3.33 | 0.000 |
| **Detection (logit-scale)** | | | | |
| intercept | −1.02 | −1.11 | −0.92 | 0.000 |
| year (2018) | 1.21 | 1.07 | 1.35 | 0.000 |
| year (2023) | 1.09 | 0.90 | 1.29 | 0.000 |
| day of year | −0.05 | −0.06 | −0.05 | 0.000 |
| elevation variation | 0.06 | 0.05 | 0.07 | 0.000 |
| year (2018):day of year | 0.03 | 0.02 | 0.03 | 0.000 |
| year (2023):day of year | 0.01 | 0.00 | 0.02 | 0.042 |

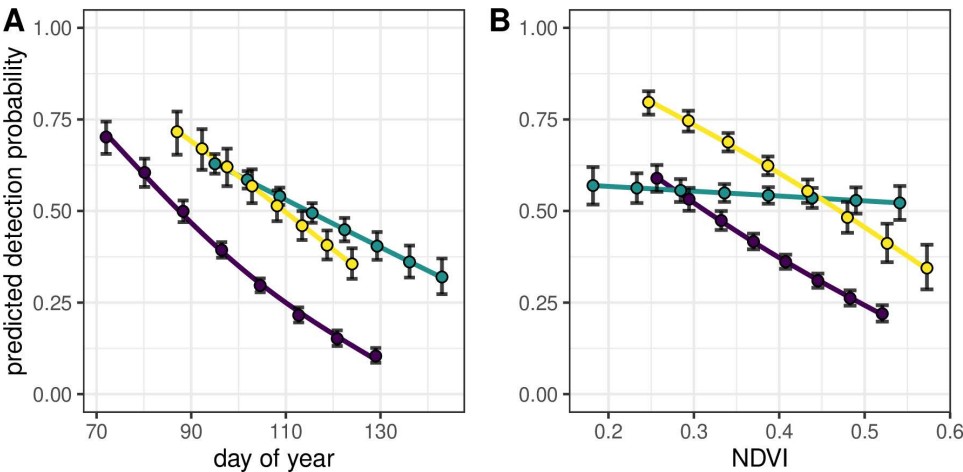

**Fig 2. Predicted detection probability of white-tailed deer (*Odocoileus virginianus*) during nocturnal spotlight surveys in Iowa, USA depending on (A) day of year and (B) Normalized Difference Vegetation Index (NDVI).** For (A) points were predicted from the N-mixture model reported in Table 1. Day of year and NDVI were correlated from $r = 0.41$–0.49 within years; we therefore fit (B) NDVI in a separate model in place of day of year for comparison. Error bars represent ± standard error and each year is a different color (dark purple = 2017, teal = 2018, yellow = 2023).

increasing NDVI and this effect varied among years (Fig 2B). However, the best NDVI model (i.e., NDVI x year) explained considerably less variation than the best day-of-year model (Δ AIC = 107).

The top model with site-level covariates on detection probability included elevation variation (S2 Table) in which sites with greater elevation variation were predicted to have greater detection probability (Table 1; Fig 3). Other site-level covariates explained variation in detection to a lesser degree but were still significant predictors (S8 Fig). Forest percent (β = 1.21, 95% confidence interval (CI) = 0.14–2.29) and hay/pasture percent (β = 1.38, CI = 0.88–1.87)

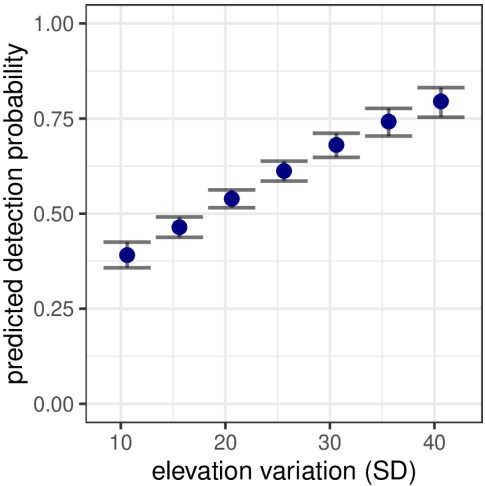

**Fig 3. Predicted detection probability of white-tailed deer (*Odocoileus virginianus*) during nocturnal spotlight surveys in Iowa, USA depending on variation in elevation.** Points were predicted from the N-mixture model reported in Table 1. Error bars represent ± standard error.

positively covaried with detection probability, whereas crop percent had a negative relationship with detection (β = −0.69, CI = −1.03− −0.35).

## Abundance

Abundance was best predicted by the percent of hay/pasture landcover, followed by crop percent and forest percent (S3 Table). Abundance increased at sites with greater hay/pasture percent (β = 2.32, CI = 1.30–3.33) and forest percent (β = 3.90, CI = 1.62–6.18), and decreased at sites with greater crop percent (β = −1.45, CI = -2.17− −0.73; Fig 4). Elevation variation did not affect abundance (β = 0.01, CI = −0.01–0.03; S9 Fig).

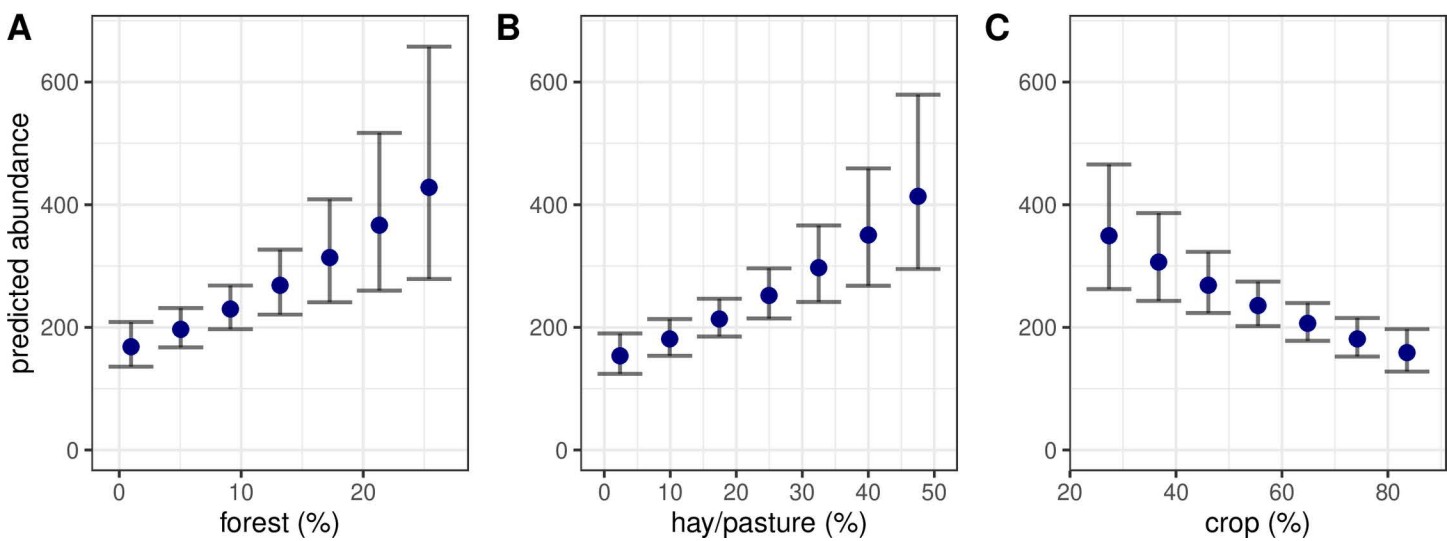

**Fig 4. Predicted abundance of white-tailed deer (*Odocoileus virginianus*) during nocturnal spotlight surveys in Iowa, USA depending on percent of different land-cover classifications.** Points were predicted from the N-mixture model reported in Table 1 (B) or its caption (A & C). Error bars represent ± standard error.

## Discussion

A common question in wildlife research is whether survey counts of animals are repeatable and provide an accurate index of abundance or whether counts vary due to variation in detectability or availability of animals. We found very high consistency in counts of white-tailed deer among temporal replicates during spring in an agriculturally dominated landscape of the Midwest, USA. We were able to explain additional variation in detection by modeling day of year and elevation variation, which should increase the accuracy of abundance estimates.

### Repeatability

We argued that confidence in survey counts could be gained from either (1) low variation in counts among temporal replicates or (2) an ability to explain variation in counts with covariates. Given eighty-six percent of the variation in survey counts was explained by site–year and only 14% of the variation was attributable to differences among temporal replicates, survey counts in our system seemed reliable because of low variation in counts among temporal replicates. Nevertheless, we were also able to explain an additional 14% of the unexplained variation in temporal replicates with observation covariates, increasing the amount of variation explained by site–year to 88% and further increasing our confidence in the usefulness of individual survey counts in our system.

### Detection probability

Day of year had the strongest effect on detection probability of any observation-level covariate. Detection probability decreased as day of year increased, which we hypothesized was caused by green-up progression through spring. We were therefore surprised that NDVI explained less variation in counts than day of year. One reason could be that cloud cover blocked parts of satellite images that we used to extract NDVI such that this measure did not perfectly reflect vegetative green-up throughout our entire dataset. Although we developed a data processing framework to reduce cloud cover bias (S1 Appendix), cloud cover in images from 2018 likely explains the shallow slope in Fig 2B. We note that the high correlation of day of year and NDVI, particularly within years, hinders our ability to parse effects between these two variables. Regardless of the mechanism causing variation in counts among temporal replicates, our results suggest white-tailed deer managers should survey early in spring prior to significant green-up to increase the proportion of the population detectable with nocturnal spotlight surveys.

Advancing green-up could reduce detection probability by increasing visual complexity on the landscape, and reduce availability if deer increase the use of forest because of new food in the form of leaf buds (*sensu* [68]). Like other ungulates, white-tailed deer tend to increase group size in late winter and break into smaller groups as spring advances [51,69–71], which could affect detectability. Group size is probably driven by changes in habitat use [69,70]. Ungulates occur in larger groups in open habitat where group vigilance may confer benefits [69,70,72,73]. As deer move to denser landcover where avoiding predator detection altogether is more feasible, deer shift to smaller groups, probably because the benefits of stealth outweigh the benefits of group vigilance. Food may be more scattered in forested habitat, dispersing deer on the landscape and indirectly reducing group size [69]. Day of year could have additional effects on deer behavior as pregnant females separate from groups in preparation for parturition (May–June in Iowa [74]; see also [51]). Lastly, agriculture fields are increasingly worked during this time period, which may disturb deer from open fields and likely reduces food potential if tilled.

Elevation variation was the only site-level covariate to have a considerable effect on detection probability. Percent forest, hay/pasture, and crops had significant associations with detection probability, but these may result from percent landcover correlations with elevation variation (Fig S5). We hypothesized that elevation variation could influence detection probability by increasing the visual complexity of the landscape and increasing the difficulty for an observer to detect deer. In contrast, we observed higher detection probability at sites with greater elevation variation. Sites with greater elevation variation likely have smaller viewsheds, which reduces the area that is sampleable (*sensu* [37,75]). Because elevation variation is likely to shorten the viewshed closer to the transect where visibility is greater, average detection probability over the area that can be sampled should be higher. Smaller viewsheds may also reduce observer fatigue [76] and focus observers on areas where deer could be detected. Topographically diverse sites have also produced reliable spotlight survey estimates of red deer in mountainous regions of Italy [39] and France [25], and aerial surveys of mule deer (*Odocoileus hemionus*) found higher detection in areas with increased ruggedness [76].

## Abundance

Abundance was higher at sites with a greater proportion of forest and hay/pasture cover and was lower at sites with greater landcover in row crop agriculture. Forests provide bedding cover and woody browse that deer rely on, and the amount of forest has been posited as the main factor limiting population size in the Midwest [77]. Corn and soybeans serve as major food sources during the summer and autumn and offer bedding cover when crops are adequately high. After crops are harvested, deer must be able to seek retreat sites that typically exist in Iowa as various types of forests (e.g., oak and hickory upland, willow and maple riparian areas) or other persistent cover in landscapes void of forests (e.g., marsh wetlands, tallgrass prairies). Thus, while it is possible that deer are negatively affected by row crop agriculture, abundance was probably lower at sites with increasing crop percent because of the correlated lack of cover at these sites.

## Comparison with other systems

The question of whether survey counts provide unbiased indices of their population is widespread among wildlife studies [14,15,26,78–80]. Yet, this question is particularly contentious for road-based nocturnal spotlight surveys of white-tailed deer (reviewed in [34]), providing a wealth of study to compare the conditions that determine whether a system can accurately be surveyed with counts of unmarked animals. Why were counts consistent and detection probability/availability predictable in our study but not others? First, we surveyed during a time period that animals were maximally exposed (i.e., at night during early spring), which led to high detection probability and availability. Population estimates from unmarked populations tend to be less accurate when detection probability is low [80–83]. Studies reported unpredictable variation in detection of deer when sampled in July and August in approximately 93% forested landcover [31,34]. Limited food resources within forest during early spring probably create a stronger pressure on deer to feed in exposed areas than summer. Second, we sampled at a time of year when deer movement and landscape use is relatively stable and sampled only on nights with similar environmental conditions. This gives staff flexibility on when they collect data, which is needed when sampling a large spatial extent, and reduces the complexity of covariate structure needed to represent the data generating process. Third, we sampled sites with ≤25% forest cover. Other studies have suggested forested landcover increases the difficulty in reliably estimating detection because visibility is impeded [27,31,34]. We actually found a slight positive association between percent forest

and detection, which may result from smaller viewsheds (discussed above). We suspect studies that find high variation in detection probability in highly forested areas are probably the result of temporal variation in animal availability because more food options exist within forest during summer. In support of this, nocturnal spotlight surveys of red deer in highly forested landcover (80% forest [25]; 73% forest [39]) find high and stable detection probabilities in early spring when quality food is limited to open landcover. Nevertheless, the ecology of deer in a warmer climate with 93% forest cover may simply not be predictable enough to accurately implement spotlight surveys [34]. In general, we suggest counts were repeatable in our system because we surveyed when deer were available to sample, landscape use was stable, and the primary influence of heterogeneity (day of year/green-up) was quantifiable. In other words, timing of surveys allowed us to minimize unmodeled heterogeneity in detection probability.

An important assumption of transect sampling is that transects are randomly located in the area that abundance is estimated [9]. Road based surveys violate this assumption and can be especially problematic if animals or habitat quality are distributed nonrandomly from roads [34,84,85]. Because our goal was to assess the temporal consistency of counts, the nonrandomness of transects was not an issue for our current study. Researchers conducting road- or trail-based surveys should consider the potential consequences of non-random transects if extrapolating density outside of the visible area. Because of their logistic ease, road-based surveys are likely to remain a widely used wildlife monitoring technique.

## Methodological considerations for assessing detection probability and availability

There is great interest in validating the reliability and assumptions of abundance estimation techniques for unmarked animals [24,43,78,80,86]. Validation is limited by not knowing the true population size [26] because a complete census is rarely feasible. Instead, researchers tend to use additional methods for comparison, which fall into two categories. First, mark-recapture/resight methods have the potential to produce the most precise estimates of detection probability because a known number of marked individuals exists in the population and detection probability can be calculated as the proportion of marked individuals that are observed on subsequent surveys [3,26]. This method assumes that marked individuals behave similarly to unmarked individuals [87] and requires substantial effort in capture and marking, which varies among taxa. Second, non-marking methods that have higher detection probability, such as thermal imaging [28,31] or aerial surveys [24,35], set a lower bound for the proportion of animals missed by lower efficacy techniques. These methods require additional equipment and expertise to conduct and still estimate abundance given imperfect detection.

We opted to temporally replicate surveys to test the repeatability of counts and assess the stability of detection probability and animal availability during an annual monitoring period (similar to [24]). Two main benefits exist using temporal replication. First, no additional training or equipment is needed. Second, this method produces repeated measures of closed populations enabling better tests of whether observation-level covariates affect detection probability or availability. Distance sampling enables tests of observation-level covariates on detection probability via the detection function but cannot separate variation in animal availability from abundance without repeated measures. Researchers should choose the method that is best suited to address the largest suspected sources of bias in their system. In reality, a combination of validation techniques would likely continue to provide bias correction and confidence in population estimates. Such investment is probably most beneficial to long-term monitoring efforts whereby the information obtained can be applied to future protocol, data analysis, and interpretation.

## Conclusions

Wildlife surveys are widely used to estimate animal abundance, despite well validated concerns that unexplained variation in detection can reduce the usefulness of survey counts. We show high consistency in survey counts among temporal replicates during a time period when animals were maximally exposed and available to sample. This high repeatability suggests that, given appropriate timing and ecological conditions, surveys can provide reliable population estimates. In particular, our results suggest deer managers should sample early in the spring green-up progression (or before) to maximize the proportion of the population that is available to sample. Systems that can validate high repeatability among replicate counts can reduce annual monitoring effort to one survey per site per year (the norm for management agencies) while maintaining confidence in population estimates. Lastly, we show how temporal replication can be a cost-effective technique to evaluate the reliability of wildlife surveys, thereby providing a more effective basis for making informed management decisions.

## Supporting information

**S1 Appendix. Data processing and cleaning methodology for a Normalized Difference Vegetation Index (NDVI).**
(DOCX)

**S1 Table. List of models dredged to determine the best fit for observation-level covariates on detection probability.** Only an intercept was fit on abundance. Models ranked by ΔAIC.
(DOCX)

**S2 Table. Table of model fits.** We selected the best fit model from S1 Table (i.e., null model here) and fit four additional models that each had one site-level covariate. Separate models were fit for each site-level covariate because of correlation among these variables (S5 Fig). Models are ranked by ΔAIC.
(DOCX)

**S3 Table. Table of model fits for abundance.** We used the top model from S2 Table (i.e., intercept model here) and built nine additional models, each represented by a row. Site-level covariates were fit in separate models because of correlation among these variables (S5 Fig). Models are ranked by ΔAIC.
(DOCX)

**S1 Fig. Correlation among observation-level covariates in the global dataset.** This figure was created in the R package *GGally*. Panels on the diagonal represent the univariate distribution for each variable. Panels up and right of the diagonal represents correlation coefficients with significance levels denoted with asterisks. Panels low and left of the diagonal represent bivariate scatterplots. Year was fit as a factor in all models and the distribution of each variable among years is shown with histograms (along bottom row) and boxplots (on right column).
(TIF)

**S2 Fig. Correlation of observation-level covariates for 2017 surveys.** This figure was created in the R package *GGally*. Panels on the diagonal represent the univariate distribution for each variable. Panels up and right of the diagonal represents correlation coefficients with significance levels denoted with asterisks. Panels low and left of the diagonal represent bivariate scatterplots.
(TIF)

**S3 Fig. Correlation of observation-level covariates for 2018 surveys.** This figure was created in the R package *GGally*. Panels on the diagonal represent the univariate distribution for each variable. Panels up and right of the diagonal represents correlation coefficients with significance levels denoted with asterisks. Panels low and left of the diagonal represent bivariate scatterplots.
(TIF)

**S4 Fig. Correlation of observation-level covariates for 2023 surveys.** This figure was created in the R package *GGally*. Panels on the diagonal represent the univariate distribution for each variable. Panels up and right of the diagonal represents correlation coefficients with significance levels denoted with asterisks. Panels low and left of the diagonal represent bivariate scatterplots.
(TIF)

**S5 Fig. Correlation of site-level covariates.** This figure was created in the R package *GGally*. Panels on the diagonal represent the univariate distribution for each variable. Panels up and right of the diagonal represents correlation coefficients with significance levels denoted with asterisks. Panels low and left of the diagonal represent bivariate scatterplots.
(TIF)

**S6 Fig. Raw counts from each survey.** Not all sites were sampled in all years. Each site was temporally replicated three times in 2017 and 2018 and twice in 2023. Points are horizontally jittered to avoid overplotting.
(TIF)

**S7 Fig. Residual counts from each survey after removing variation attributable to observation-level covariates (day of year, temperature, humidity).** We added the intercept to each point so residual values center around the raw mean. Not all sites were sampled in all years. Each site was temporally replicated three times in 2017 and 2018 and twice in 2023. Points are horizontally jittered to avoid overplotting.
(TIF)

**S8 Fig. Predicted detection probability of white-tailed deer (*Odocoileus virginianus*) during nocturnal spotlight surveys in Iowa, USA depending on percent of different land-cover classifications.** Landcover types were correlated (S5 Fig), and therefore fit in separate models. Points were predicted from the N-mixture model reported in Table 1. Error bars represent ± standard error.
(TIF)

**S9 Fig. Predicted abundance of white-tailed deer (*Odocoileus virginianus*) during nocturnal spotlight surveys in Iowa, USA depending on variation in elevation.** Error bars represent ± standard error.
(TIF)

**S1 Data. Archived dataset.**
(CSV)

## Acknowledgments

We thank all the staff and volunteers that conducted spotlight surveys. We thank Bob Klaver and Chris Ensminger for discussion that improved the paper. Reviews from Garth Mowat and three anonymous reviewers improved the paper.

## Author contributions

**Conceptualization:** David M. Delaney, Tyler M. Harms, Jonathan P. Harris, Dan J. Kaminski, Jace R. Elliott, Stephen J. Dinsmore.

**Data curation:** Tyler M. Harms, Dan J. Kaminski, Jace R. Elliott.

**Formal analysis:** David M. Delaney.

**Methodology:** David M. Delaney, Tyler M. Harms, Jonathan P. Harris, Dan J. Kaminski, Jace R. Elliott, Stephen J. Dinsmore.

**Project administration:** Tyler M. Harms, Stephen J. Dinsmore.

**Supervision:** Tyler M. Harms, Stephen J. Dinsmore.

**Validation:** Jonathan P. Harris.

**Visualization:** Jonathan P. Harris.

**Writing – original draft:** David M. Delaney.

**Writing – review & editing:** David M. Delaney, Tyler M. Harms, Jonathan P. Harris, Dan J. Kaminski, Jace R. Elliott, Stephen J. Dinsmore.

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
