## [Decision Letter · Decision Letter 0]

23 Jan 2025

PONE-D-24-57166Repeatability of spotlight surveys for wildlife abundance estimationPLOS ONE

Dear Dr. Delaney,

Thank you for submitting your manuscript to PLOS ONE. After careful consideration, we feel that it has merit but does not fully meet PLOS ONE’s publication criteria as it currently stands. Therefore, we invite you to submit a revised version of the manuscript that addresses the points raised during the review process.

We look forward to receiving your revised manuscript.

Kind regards,

Stephen M. Rich, MS, PhD

Academic Editor

PLOS ONE

4. We note that there is identifying data in the Supporting Information file < archived_data.xlsx>. Due to the inclusion of these potentially identifying data, we have removed this file from your file inventory. Prior to sharing human research participant data, authors should consult with an ethics committee to ensure data are shared in accordance with participant consent and all applicable local laws.

-Location data

Please remove or anonymize all personal information \, ensure that the data shared are in accordance with participant consent, and re-upload a fully anonymized data set. Please note that spreadsheet columns with personal information must be removed and not hidden as all hidden columns will appear in the published file.

Reviewers' comments:

Reviewer's Responses to Questions

**Comments to the Author**

1. Is the manuscript technically sound, and do the data support the conclusions?

Reviewer #1: Yes

Reviewer #2: Yes

2. Has the statistical analysis been performed appropriately and rigorously? 

Reviewer #1: Yes

Reviewer #2: Yes

3. Have the authors made all data underlying the findings in their manuscript fully available?

Reviewer #1: Yes

Reviewer #2: Yes

4. Is the manuscript presented in an intelligible fashion and written in standard English?

Reviewer #1: Yes

Reviewer #2: Yes

5. Review Comments to the Author

Reviewer #1: I think this is a very useful paper to add to the literature to evaluate index methods of animal abundance. The paper is well written and clear, I have made a few grammatical suggestions on the ms using the Adobe Editor tools. They are all minor. I have a few larger questions that I believe the authors should consider before finalizing the paper.

1. I think the title undersells both the objectives of the paper and what you did, you looked at more than just repeatability.

2. Line 62-“For systems with repeatable counts and predictable detection probability, researchers can reduce visits to one sample per site and reallocate effort elsewhere.” Are you assuming then that if the count is repeatable that availability bias is low? In the Introduction you very clearly lay out the 2 main sources of bias to a count method and mention that it is hard to identify the two, but I do not recall any further specific discussion of availability after that. I think that in your suggestion that sightability is very high in your open system you are assuming that availability is high? But you also suggest that viewing distance changes with elevation variation, could that affect availability?

3. Why does topographic variation positively affect detection? In theory if you include a variable in a model and it generates fit opposite to expectation then you should drop that variable from the model because the fit is spurious because you cannot explain it. Are you sure that the variation in elevation is not positively affecting abundance and that you are mistaking one factor for the other? Maybe there is more food in places with topographic variation?

4. In the Introduction or Methods section please add a sentence explaining to the reader how a mixture model can estimate both abundance and detection from simple count data. This is not an intuitive idea but is important to your paper and especially important to my question above about identifiability.

5. Are there large predators in your system that may affect nocturnal choice of habitat by deer, and therefore availability?

6. In line 278 can you conclude that temporal variation is only partially related to green-up and equally influenced by other factors? If so, then I suggest you add a closing sentence stating that clearly. This is important for future sampling design.

7. Mgmt Recommendations: here can you not describe or suggest the methodological improvements you suggest should be applied to these surveys? What is the preferred survey window? What covariates should be collected, which can be dropped? Which model should the practitioners use to index or estimate deer abundance. Can you not now estimate abundance using the mixture model and avoid the index all together? Can they apply the model to the historical data and expect better retrospective inference?

8. I am thinking about the paper from a statistical point of view because I cannot help but think you learned more than you have presented in your Mgmt Implications section. I am rambling a bit below while I try to put my finger on what you have overlooked.

9. Step 1 examined precision and you were quite lucky in that it is high. What would you have recommended if it was low? Given it is high then can you word your recommendation in lines 382-385 more directly-only one count is needed if it is done between these dates?

10. Step 2 examined detection success and given precision was high this should be the main signal in the data? Or can you say, based on the strength of your abundance model how much abundance may have confounded detection? Here your hypothesis that seasonal timing effects detection was supported while the green-up hypothesis was less supported. Again, given the strength of those results, I think you can make a recommendation about actual survey timing in your system, and, for the benefit of others, tell use about the typical weather and deer social behaviour during that time. Can you say why the seasonal effect varied among years or, is that likely just due to variation in the data among years (ie random)? As above, is the positive association with pastureland identifiable as a detection effect, or could it be confounded with an influence on abundance? Regardless of your response, should this relationship not be mentioned in the Mgmt Implications section? My thought is to leave the habitat variable out of the detection model because you already know it influences abundance.

11. Your Step 3 examined influences on abundance and here availability may be the largest influence because precision is high, and you have accounted for a number of factors influencing detection probability. As I mentioned above, it seems to me you now have the tools to estimate abundance and again, whether you agree with me or not, I think you need to mention in the Mgmt Implications section that you examined abundance, while controlling for precision (sort of), and detection, and found abundance was related to habitat type. Which is something I suspect your surveyors have been seeing for years.

Garth Mowat

University of British Columbia

Reviewer #2: This is a clear and concise paper on the repeatability of count data and methods for assessing it. I have minor line comments, but the more major of those are asking why the mixed model was fit using a Gaussian distribution and a recommendation to rewrite the N-mixture model section because it was difficult to follow which covariates were used when and why.

26-28: Not sure what this means. If you’re doing single-visit sampling how can you use temporal replication to assess them?

141-143: What is the GGally package actually calculating? Why couldn’t you just look at the correlation coefficient?

145: Does this mean every site-year combination was treated as a random effect?

146: Need to indicate if high values of this fraction indicate high or low repeatability.

148: Why Gaussian and not Poisson?

154-155: If you are using a negative binomial here, no reason not to use Poisson/negative binomial in the mixed model.

161: “fit” instead of “run”

163: Define uninformative parameter

164-170: Is this for the N mixture model as well? Why not include NDVI originally?

171-181: Does this mean elevation either appeared in the model for abundance OR detection, but no model was fit with them together? Just having trouble following the sequence of models.

178-180: It seems like the fact that you included transect length as a covariate is information that should come earlier. I would recommend rewriting the N-mixture model section quite a bit, it’s hard to follow which covariates are appearing when and why. Covariates that are in all the models regardless should be introduced first, then the core set of covariates you are testing, then the additional covariates and which models they are going into.

184: This R is not a correlation coefficient correct? It’s proportion of variance explained by site-year combination? Still having a hard time thinking about what a high value means here.

Discussion: Given you found that abundance and detection probability were influenced by the same covariates, do you have confounding in the two processes? Can you actually separate abundance from detection/availability?

6. PLOS authors have the option to publish the peer review history of their article (what does this mean? ). If published, this will include your full peer review and any attached files.

**Do you want your identity to be public for this peer review?** For information about this choice, including consent withdrawal, please see our Privacy Policy .

Reviewer #1: **Yes: ** Garth Mowat

Reviewer #2: No

---

## [Editor Report · Decision Letter 1]

10 Mar 2025

Repeatability of wildlife surveys for estimating abundance: A method to assess the consistency of detection probability and animal availability

PONE-D-24-57166R1

Dear Dr. Delaney,

We’re pleased to inform you that your manuscript has been judged scientifically suitable for publication and will be formally accepted for publication once it meets all outstanding technical requirements.

Kind regards,

Stephen M. Rich, MS, PhD

Academic Editor

PLOS ONE
---

## [Editor Report · Acceptance letter]

PONE-D-24-57166R1

PLOS ONE

Dear Dr. Delaney,

I'm pleased to inform you that your manuscript has been deemed suitable for publication in PLOS ONE. Congratulations! Your manuscript is now being handed over to our production team.

Kind regards,

on behalf of

Dr. Stephen M. Rich

Academic Editor

PLOS ONE